# Role of PatAB Transporter in Efflux of Levofloxacin in *Streptococcus pneumoniae*

**DOI:** 10.3390/antibiotics11121837

**Published:** 2022-12-17

**Authors:** Mónica Amblar, Ángel Zaballos, Adela G de la Campa

**Affiliations:** 1Centro Nacional de Microbiología, Instituto de Salud Carlos III, Ctra Majadahonda-Pozuelo Km 2.200, Majadahonda, 28220 Madrid, Spain; 2Unidades Centrales Científico Técnicas, Instituto de Salud Carlos III, Ctra Majadahonda-Pozuelo Km 2.200, Majadahonda, 28220 Madrid, Spain; 3Presidencia, Consejo Superior de Investigaciones Científicas, 28006 Madrid, Spain

**Keywords:** *Streptococcus pneumoniae*, PatAB, fluoroquinolone-resistance, efflux-pump, levofloxacin–efflux

## Abstract

PatAB is an ABC bacterial transporter that facilitates the export of antibiotics and dyes. The overexpression of *patAB* genes conferring efflux-mediated fluoroquinolone resistance has been observed in several laboratory strains and clinical isolates of *Streptococcus pneumoniae*. Using transformation and whole-genome sequencing, we characterized the fluoroquinolone-resistance mechanism of one *S. pneumoniae* clinical isolate without mutations in the DNA topoisomerase genes. We identified the PatAB fluoroquinolone efflux-pump as the mechanism conferring a low-level resistance to ciprofloxacin (8 µg/mL) and levofloxacin (4 µg/mL). Genetic transformation experiments with different amplimers revealed that the entire *patA* plus the 5’-terminus of *patB* are required for levofloxacin–efflux. By contrast, only the upstream region of the *patAB* operon, plus the region coding the N-terminus of PatA containing the G39D, T43A, V48A and D100N amino acid changes, are sufficient to confer a ciprofloxacin–efflux phenotype, thus suggesting differences between fluoroquinolones in their binding and/or translocation pathways. In addition, we identified a novel single mutation responsible for the constitutive and ciprofloxacin-inducible upregulation of *patAB*. This mutation is predicted to destabilize the putative rho-independent transcriptional terminator located upstream of *patA*, increasing transcription of downstream genes. This is the first report demonstrating the role of the PatAB transporter in levofloxacin–efflux in a pneumoccocal clinical isolate.

## 1. Introduction

*Streptococcus pneumoniae* (SPN) is a major cause of morbidity and mortality worldwide, and the main etiological agent of pneumonia, meningitis, septicemia, and acute otitis media [1]. Antibiotic resistance to traditionally effective drugs, such as β-lactams [2] or macrolides [3], is an increasing problem to treat pneumococcal infections in many countries [2,3,4,5]. The so-called respiratory fluoroquinolones (FQs), such as levofloxacin (LVX) and moxifloxacin, are active in the treatment of pneumococcal pneumonia and are currently used in adult patients [6,7,8]. However, the emergence of the FQ-resistant pneumococci is a growing concern and treatment failure may occur due to infection with a FQ-resistant strain or to the emergence of resistance during therapy [9]. The intracellular targets of FQs are the DNA topoisomerase enzymes topoisomerase IV and gyrase, which solve topological problems associated with replication, transcription and recombination [10]. Resistance to FQs is mainly due to alterations in the quinolone-resistance determining regions of their target topoisomerases, which are acquired either by point mutations, or by interspecific recombination with genetically related streptococci [11,12]. Low-level resistance to FQs can also be the result of active efflux. Although this mechanism usually leads to lower MICs than mutations in DNA topoisomerases [13,14,15], several studies suggest that efflux pumps are a requirement for the selection of FQ resistance in SPN [16].

Three different FQ-efflux pumps have been identified in SPN so far: PmrA, DinF and PatAB. PmrA (member of the major facilitator superfamily) was the first efflux pump shown to confer resistance to ciprofloxacin (CPX) and norfloxacin in SPN [17] and the deletion of the multi-antimicrobial extrusion DinF caused susceptibility to several FQs [18]. However, the only system shown to confer FQ-resistance in clinical isolates was PatAB. This pump belongs to the ATP-binding cassette superfamily (ABC-transporter) and is composed of two transporters encoded by *patA* and *patB* genes [19,20]. ABC-transporters are integral membrane proteins that actively transport chemically diverse substrates across the lipid bilayer [21] and include clinically relevant examples such as the human protein MDR1 (also known as P-glycoprotein) and MRP1, which confer resistance to anticancer cytotoxic compounds [22]. PatA and PatB proteins form a heterodimeric transporter, where both proteins have a transmembrane domain and an ATP binding domain [23].

Previous studies demonstrated that the inactivation of the PatAB transporter was associated with a hypersusceptibility to norfloxacin, CPX, acriflavine and ethidium bromide in SPN [20,24], and its homolog SatAB in *Streptococcus suis* was also involved in CPX efflux [25]. The constitutive overexpression of *patA* and *patB* genes has been observed in SPN laboratory strains and FQ-resistant clinical isolates [24]. Such an overexpression has been associated to mutations in the upstream region of *patA*, which affect formation of a rho-independent terminator [26,27]. In addition, it has been demonstrated that exposure to FQs and DNA damage strongly induces the expression of *patA* and *patB* [19,28]. Another regulator of the *patAB* expression in *S. suis* is the two-component system, CiaRH [29].

In this report, we study the mechanisms of FQ-resistance of the multidrug resistant pneumococcal isolate SPN1852. This isolate was obtained during 2006 in a study of the prevalence of FQ-resistance performed with 4215 pneumococci from 110 hospitals throughout Spain [30]. SPN1852 showed a low-level resistance to CPX and LVX. Here, we demonstrate that this low-level resistance is due to FQ efflux, since it is susceptible to reserpine (Res). Through genetic experiments and whole genome sequencing, we demonstrate that PatAB is responsible for this efflux phenotype. A novel single mutation in the upstream region of *patAB*, plus the region coding the N-terminus of PatA containing four amino acid changes, is sufficient to increase transcription and confer CPX-efflux in a sensitive strain. However, four additional amino acid changes in the N-terminus of PatB were required for LVX-efflux. Although previous studies have described the pump efflux associated to a resistance to CPX and norfloxacin in Streptococci [17,24,25,31,32], to our knowledge, this is the first report establishing the role of PatAB in levofloxacin–efflux in a pneumoccocal clinical isolate.

## 2. Results

### 2.1. Clinical Isolate 1852 Exhibits a Fluoroquinolone Efflux Phenotype

Clinical isolate SPN1852 was previously identified as *S. pneumoniae* (SPN) by standard methods, and the serotype 19A was determined by the Quellung reaction [30]. To better characterize this isolate, we performed multilocus-sequence-typing (MLST) following the seven housekeeping genes scheme included in SPN MLST [33]. Sequence analysis demonstrated a new allelic profile (*aroE*: 59; *gdh:* 473; *gki:* 666; *recP*: 490; *spi*: 313; *xpt*: 918 and *ddl*: 702) not described in the MLST database and, therefore, a new ST number was assigned, ST-17858.

The susceptibility of SPN1852 to a range of antibiotics was analyzed by agar-plate dilution, showing resistance to penicillin (0.5 µg/mL), tetracycline (64 µg/mL), and erythromycin (32 µg/mL). Regarding fluoroquinolones (FQs), SPN1852 showed a low-level resistance to ciprofloxacin (CPX) and levofloxacin (LVX) (MICs of 8 and 4 µg/mL, respectively), but was susceptible to moxifloxacin. This isolate did not carry mutations in the quinolone-resistance determining regions of their DNA topoisomerases [30]. To investigate whether its low-level FQ-resistance was associated to an efflux phenotype, we analyzed the susceptibility of the isolate to CPX and LVX in the presence or absence of the efflux-pump inhibitor Res and compared it to those of the susceptible laboratory strain, R6. As shown in Table 1, the MICs for the R6 strain of each antibiotic were equivalent, independently of the Res addition. By contrast, SPN1852 showed 4- (LVX) or 16-fold (CPX) higher MICs in the absence than in the presence of Res (Table 1). Since the concentration of Res used (20 µg/mL) was 3-fold lower than SPN MIC, no effect on growth was expected [34]. These results suggested the existence of a Res-sensitive efflux pump in SPN1852 able to extrude both CPX and LVX out of the cell.

To identify the genes linked to this efflux-phenotype, strain R6 was transformed with chromosomal DNA of SPN1852, as described in the methods. Transformed cells were plated onto blood–agar plates containing 0.5 µg/µL of LVX. It yielded 88 CFUs (transformation efficiency 8.8 × 10^3^ CFUs per mL and per µg). Susceptibility to LVX and CPX of seven of these transformants was assayed with or without Res and all of them exhibited an identical efflux phenotype than SPN1852 for both drugs (Table 1).

To confirm the presence of a CPX-efflux phenotype, the in vivo intracellular accumulation of CPX was measured in SPN1852, H4 and R6 strains, with and without Res. As expected, the accumulation of CPX in R6 was similar either in the presence or in the absence of Res (Figure 1). By contrast, in SPN1852 and transformant H4 a clear efflux phenotype was observed. The levels of intracellular CPX without Res in both strains were significantly lower than in R6, with reductions ranging between 2.2- and 5.2-fold. However, when Res was added to the culture, the levels of CPX were 1.9- to 4.0-fold higher than in non-treated cells (Figure 1), reaching a similar CPX accumulation than in R6. These results suggest the activity of the Res-sensitive PatAB transporter in pumping out CPX.

### 2.2. PatAB Operon Is Responsible for FQ-Efflux in SPN1852

To identify mutations causing the FQ-efflux phenotype, whole genomes from transformants H4 and B9 were sequenced using Illumina technology. Sequence reads were aligned to the genome sequence of SPN R6 (accession number NC_003098.1) and variants identified with the GS Reference Mapper tool of the Newbler package. H4 transformant showed 706 nucleotide changes with respect to R6, while B9 showed 608 changes. After subtracting those changes present in our laboratory strain R6 [32], the changes attributed to the recombination event were 681 and 580 for H4 and B9, respectively (Appendix A). In B9, all changes were located in a 12765-bp region, which covers from *gltX* to *argS* loci (Figure 2a), except for two point mutations located in *SPR_RS02055* and in *galU*. In H4, the recombination occurred in two different regions, one located in a 5404-bp region covering *fusA* and *polC* genes containing 256 changes, and another region of 9152-bp with 423 changes, located in equivalent positions to the recombination region found at B9, covering *pgi* to *argR* (Figure 2a). The two remaining nucleotide changes corresponded to point mutations in *SPR_RS01895* and *SPR_RS05495*. Therefore, H4 and B9 showed a common recombination region including genes *pgi*-*guaA*-*patB*-*patA*-*hexA*-*argR*. A similar region containing *guaA*-*patB*-*patA*-*hexA* was previously obtained in genetic transformation experiments performed with chromosomal DNA from a *Streptococcus pseudopneumoniae* (SPS) resistant clinical isolates [32], and *patA* and *patB* genes have been linked to CPX-efflux phenotypes in SPN and SPS [24,32]. Therefore, we ruled out the possible involvement of the remaining changes in the efflux phenotype and focused on the *patAB* operon.

We determined by the Sanger method the sequence of a DNA fragment including the *patA*, *patB* and the *hexA*-*patA* intergenic region (IGR) in SPN1852 and transformants B9 and H4, and compared them to R6 and to the *S. pseudopneumoniae* type strain (SPST). Eight changes were found with respect to the Illumina method: Seven additional variations and one discrepancy (Appendix A). Nevertheless, both transformants, B9 and H4, exhibited identical nucleotide sequences as the isolate SPN1852. This isolate demonstrated a nucleotide divergence lower than 7% when compared to R6, with 4.8% in the *hexA*-*patA* IGR, 5.8% in *patA* and 6.3% in *patB* (Figure 2). A similar divergence was observed between SPN1852 and SPST: 2.7% in the *hexA*-*patA* IGR, 4.8% in *patA* and 6.7% in *patB*. In addition, SPN1852 and SPST lacks a 773-bp fragment in the *patA-patB* IGR (IGR_773_), which is present in R6. Instead, the *patA-patB* IGR of SPN1852 and SPS^T^ consist in only one nucleotide (IGR_1_), suggesting that a recombination between SPN and SPS may have occurred in nature.

Regarding protein sequences, a multiple sequence alignment of PatA and PatB revealed a high sequence identity between SPN1852, R6 and SPST (Figure 3). In SPN1852, the PatA was longer than in R6 with four additional residues at the C-terminus and showed eight amino acid substitutions with respect to R6. The same differences were found at SPST PatA, which also demonstrated three additional amino acid changes. In the case of PatB, SPN1852 differed in 15 residues from R6. Most of the amino acid changes (11 out of 15) were also present in the SPST PatB, which showed 22 changes when compared to R6. Like other ABC-transporters, PatA and PatB contain two different domains (as predicted by InterPro database; https://www.ebi.ac.uk/interpro/ (accessed on 22 August 2022)): a highly conserved ATP-binding domain at the C-terminus, and a less conserved transmembrane domain at the N-terminus (Figure 4), in which six putative transmembrane alpha-helices (TMH) are predicted (using https://dtu.biolib.com/DeepTMHMM (accessed on 14 November 2022)) (Figure 3). None of the changes exhibited by SPN1852 PatA were located in the predicted transmembrane regions, while in PatB, four out of the six TMH showed amino acid changes with respect to the R6 protein. Nevertheless, these changes did not affect the transmembrane prediction and the same TMH were predicted for the SPN1852 and SPST PatB.

### 2.3. Some Changes in PatA and PatB Are Dispensable for CPX- but Not for LVX- Efflux

To explore the contribution of PatA and PatB to the CPX and LVX-efflux phenotype observed in SPN1852, different PCR fragments, containing distinct regions of the *patAB* operon, were amplified from its chromosomal DNA (Figure 4a). These fragments were used as donor DNAs in transformation experiments using R6 as receptor, and transformants were selected with 1 µg/µL of CPX. Only PCRs 7, 9, 10, and 11, which include at least the upstream region of *patAB* and most of *patA,* rendered CPX-resistant transformants. PCR7 spans the whole *patAB* operon, while the shorter PCR9 contained only the first 299 nucleotides of *patB*. PCR10 includes the *patA*-IGR_1_-24 nucleotides of *patB*, and PCR11 carries the *patA* without 53 nucleotides of its 3’-end. Despite their differences, the transformation efficiencies obtained with these PCRs were similar and ranged from 1.1 to 4.9 × 10^3^ CFUs per mL and per µg of DNA. No transformants were obtained with the other PCRs, not even with PCR6, containing the upstream region and the first 172 nucleotides of SPN1852 *patA*, nor with PCR5, which spans the whole *patA* and *patB* genes but lacks the upstream region. These results suggest that the *patAB* upstream region of SPN1852 is necessary but not sufficient to provide a CPX-resistance phenotype.

Six colonies of each transformation were selected for DNA sequencing, and their MICs of CPX and LVX with or without Res were determined (Figure 4b). The six transformants obtained with PCR7 exhibited the same CPX and LVX efflux phenotype, showing MICs similar to those of SPN1852. All of them harbor a *patAB* operon with a nucleotide sequence identical to SPN1852, except for *patB*, which differed between transformants. The shortest sequence of SPN1852 *patB* found corresponded to the first 252 nt, which include the four amino acid changes F16L, F30L, S33A and N55D with respect to R6 (PCR 7 transformant, clones 1 and 6). Regarding the transformants obtained with PCRs 9, 10 and 11, all of them shared with SPN1852 the upstream region and different parts of *patA*, but not *patB* or the IGR between both genes, which was identical to that of R6. All transformants showed a CPX efflux phenotype (with 4-fold higher MICs in the presence of Res than in its absence) but not to LVX, and the MICs were 2- to 4-fold lower than that obtained with the PCR7 transformants. These results suggest that for LVX-efflux, both the whole PatA and the N-terminus of PatB of SPN1852, including substitutions F16L, F30L, S33A and N55D, are necessary. By contrast, for the CPX-efflux, only the upstream region of *patAB* from SPN1852 and the region coding amino acid substitutions G39D, T43A, V48A and D100N in PatA are required.

### 2.4. A Single Mutation in the Stem-Loop Structure Upstream PatA Increased PatAB Expression

To investigate whether the loss of LVX-resistance was associated to differences in transcript levels, we measured the expression of *patA* and *patB* in one representative clone of transformations with PCR7 (PCR7_6), PCR9 (PCR9_6) and PCR11 (PCR11_2) and compared them with R6 and H4. It has been demonstrated that CPX induces the transcription of *patAB* in clinical isolates [19,28]. Therefore, we analyzed the transcript levels of *patA* and *patB* after 1 h of incubation with 30 × their MICs of CPX (Figure 5). Strain H4 showed higher levels of expression of both *patA* and *patB* than R6, with a 2.8- and 3.2-fold increment, respectively. Moreover, unlike R6, the treatment with CPX increased the expression levels of *patA* and *patB* up to 5.3 and 6.1 times, respectively. The three PCR transformants exhibited similar transcript levels to H4 at all conditions tested, showing a constitutive increment of *patA* and *patB* expression of about 2.5 times, and a CPX-inducible increment ranging between 4.2 and 6.1 times. Moreover, the levels of *patB* in transformant 7_6, containing the short IGR_1_, was equivalent to those in transformants 9_6 and 11_2, harboring the long IGR_773_, indicating that the expression of *patB* is not affected by the length of the IGR.

Previous studies demonstrated that the CPX resistance in SPS and SPN was associated to the IGR *hexA*-*patA*, and postulated the formation of a stem-loop (SL) structure in this region responsible for CPX-induction of *patAB* expression [26,27,32]. Our transformation experiments also evidenced that the *patAB* upstream region of SPN1852 is essential for CPX-resistance. This isolate showed six mutations with respect to R6 in the *hexA-patA* IGR (Figure 6a). Two of them are located upstream the *patA* promoter, three are in the interspace between −35 and −10 promoter sequences, and one at position +31 from the transcriptional start site (+1). Sequence of the *hexA-patA* IGR in the three CPX-resistant transformants varied from total (7_6) to partial (9_6 and 11_2) identity to the sequence of SPN1852, but *patAB* transcription was similar in all of them regardless of the differences. Substitution of G (+31) by A was the only mutation present in all transformants. This mutation is located in the putative SL and may affects its stability. We used the RNA-fold program to predict the RNA structure and determine the folding free energy. The G(+ 31)→A mutation was predicted to destabilize the SL by increasing its free energy from −13.4 kcal/mol in R6 to −9.5 kcal/mol in SPN1852 (Figure 6b), which might ease the transcription of downstream genes.

## 3. Discussion

The PatAB multidrug efflux pump is involved in resistance to ciprofloxacin (CPX) and norfloxacin in clinical isolates of *S. pneumoniae* (SPN) [19,20,24,28,32], but its contribution to levofloxacin (LVX)–efflux phenotype had not yet been established. In this study, we demonstrated that the *patAB* operon of SPN1852 isolate is responsible for its reserpine (Res)-sensitive CPX and LVX low-level resistance (8 and 4 µg/mL, respectively). A DNA fragment amplified from SPN1852, containing *patA* and *patB* and their upstream region, was enough to transform the susceptible R6 strain into a CPX- and LVX-resistant strain with MICs similar to SPN1852. We identify a novel single mutation in the upstream *patAB* region causing constitutive and inducible upregulation of *patA* and *patB* genes. Our genetic studies demonstrate that the upstream region of *patAB* operon is essential for fluoroquinolone (FQ) resistance, since transformants resistant to CPX could only be selected with PCR fragments including this region. SPN1852 shows six mutations compared to R6 in this region (Figure 6b), but only the G(+31)→A substitution appears to be responsible for the FQ-resistance phenotype. This mutation destabilizes the predicted stem-loop (SL) structure upstream *patA* and is accompanied by a higher constitutive (2.1- to 2.8-fold increment) and CPX-inducible (4.2- to 6.1-fold increment) expression of the operon genes. The putative SL is followed by a run of thymine (T) residues and has been proposed to function as an attenuator [26,27]. Disruption of SL is predicted to reduce polymerase pausing and promote transcription to continue through downstream genes. In fact, additional single mutations previously identified were predicted to destabilized the SL, causing a constitutive upregulation of *patAB* [26,27,35,36]. Similarly, different insertions or deletions observed in two FQ-resistant isolates of *S. pseudopneumoniae*, were shown to enhance the CPX-induction of expression due to the reduced stability of the putative SL [32]. Altogether, these results provide a strong support for the hypothesis that the terminator structure upstream *patA* is important for the suppression of *patAB* expression, whose unfolding is necessary for transcription. Moreover, previous transcriptomic studies revealed that *patAB* genes are located in a chromosomal domain that is upregulated by global DNA changes of supercoiling [37,38], and the transcriptional response to CPX involves the upregulation of *patAB* [19]. It was previously proposed that CPX induces local changes in supercoiling, reducing the SL stability and allowing transcriptional read-through into *patA* [32]. On these basis, higher free energy of the SL due to mutations such as G(+31)→A, may ease disruption of the structure by DNA supercoiling, enhancing CPX-induction of expression.

Moreover, our studies revealed different requirements for LVX- and CPX-resistance. The shortest sequence from SPN1852 required for the LVX-efflux phenotype was carried by transformants 7_1 and 7_6. They contained the *hexA*-*patA* intergenic region (IGR), the entire *patA* and the first 252 nucleotides of *patB*, which encodes amino acid changes F16L, F30L, S33A, and N55D (L-L-A-D sequence). By contrast, only the *hexA*-*patA* IGR and the G39D, T43A, V48A and D100N amino acid changes of PatA (D-A-A-N motif) were sufficient to confer CPX-efflux phenotype. Other changes in PatA such as I318V, D382E, N428D, N536D, or the terminal GKEE residues, did not make any differences in FQ-resistance.

In addition to the L-L-A-D PatB sequence, transformants 7_1 and 7_6 contained the 1-nucleotide *patA-patB* IGR from SPN1852 (IGR_1_), instead of the longer 773-nucleotides IGR from R6 (IGR_773_). A role of this IGR in *patB* transcription could be expected. However, *patB* transcript levels in transformant 7_6, harboring the short IGR_1_, was the same as in transformants 9_6 and 11_2, which harbor the longer IGR_773_, and all of them showed similar levels to that of the chromosomal transformant, H4. This means that the differences in LVX-efflux may rely on the amino acid changes in PatB. Residues F16, F30 and S33 of R6 PatB are located in the predicted transmembrane helix 7 (TMH7), while residue N55 is part of the predicted extracellular loop that connects TMH7 and TMH8. Substitutions F16L, F30L and S33A in the SPN1852 protein can be considered conservative, and indeed do not affect the TMH prediction. However, residues from all TMHs are believed to contribute to the surface of drug translocation pathway across the membrane [39,40,41,42], and mutations in these residues could alter this process. In fact, the aromatic F has about a 12% larger hydrophobic surface area than L [43], and substitution of L by F in class A amphipathic helical peptide affected its interaction with lipids [44]. Moreover, substitution of the polar S33 by the non-polar A could also affect drug binding. Similarly, the role of extracellular connecting loops on drug recognition is still unknown, but the substitution of a positively by a negatively charged residue (N55D) could have an effect. Overall, our results demonstrate that differences in LVX binding and/or translocation pathway are due to the mutated residues.

In addition, transformants obtained with PCR9, 10 and 11 were resistant to CPX but not to LVX, but still exhibited the CPX-efflux phenotype. These transformants only acquired the *hexA*-*patA* IGR and different parts of the *patA* from SPN1852. The residual CPX-resistance phenotype might be due exclusively to the higher levels of expression caused by the mutation in the SL. However, a role of the D-A-A-N amino acid motif of PatA cannot be ruled out, since no resistant colonies lacking this sequence were obtained. Residues D-A-A of SPN1852 PatA are located in the extracellular loop connecting TMH1 and TMH2, while N is located in the intracellular loop connecting TMH2 and TMH3. It is known that ATP binding and hydrolysis in the nucleotide-binding domains generates conformational changes that are transmitted to the TM domains through non-covalent interactions at the shared interface, and that this transmission is mechanistically crucial. It has been demonstrated that the intracellular loop connecting TMH2 and TMH3 contributes to this interface in the ABC transporter Sav18665 of *Staphylococcus aureus* [45] and provides crucial contacts with the nucleotide-binding domain in the human MDR1 protein of the glycoprotein-P multidrug efflux pump [46]. These results suggest that the D100N change could affect the crosstalk between the nucleotide-binding and transmembrane domains in the pneumococcal PatAB, thus contributing to CPX-efflux. Consistent with this, when we transformed with PCR6, which includes the D-A-A amino acid sequence in PatA but no N100, we were not able to obtain transformants resistant to CPX, thus reinforcing the possible role of this residue.

Given the spread of resistance to penicillin and macrolides in SPN, the FQs LVX and moxifloxacin are used nowadays in the treatment of pneumococcal infections. Although FQ-resistance in this bacterium is not currently a problem, increase in resistance may occur in tandem with their increased use. The PatAB efflux pump confers low-level resistance to LVX, but its presence in clinical isolates could favor the appearance of topoisomerase mutations that confer high-level, clinically relevant, resistance [16]. Knowledge of the molecular bases of PatAB efflux pump, CPX and LVX efflux requirements and *patAB* transcription regulation is important for antibiotic therapy.

## 4. Materials and Methods

### 4.1. Bacterial Strains, Growth Condition and Transformation

Clinical isolate SPN1852 was previously obtained from the broncho-alveolar lavage of an adult patient with pneumonia and identified by standard methods [30]. Pneumococci were grown as static cultures in a casein hydrolase-based medium (AGCH) supplemented with 0.3% sucrose and 0.2% yeast extract (A + SY). Transformation of *S. pneumoniae* (SPN) R6 was performed, as previously described [32] using either 0.1 μg/mL of SPN1852 chromosome or 0.4 μg/mL of PCR fragments. Genomic transformants were plated in blood–agar plates containing 0.5 μg/mL LVX and colonies were counted after 24 h of growth at 37 °C in a 5% CO_2_ atmosphere. PCR transformants were plated in A + SY media plates with 1% agar containing 1 to 2 µg/mL of ciprofloxacin (CPX) and transformation was further verified by DNA sequencing.

### 4.2. MLST, Multilocus Sequence Analysis (MLSA)

Multilocus sequence typing (MLST) of AC1852 was carried out, as previously described [47]. Tentative allele number analysis was performed by comparing the sequences obtained to those in the pneumococcal MLST database using the PubMLST.org website [33]. The allelic combination obtained was assigned a new St number ST-17858.

### 4.3. Whole Genome Sequencing

R6 transformants H4 and B9 were subjected to whole genome sequencing. Genomic DNA samples were prepared from mid-log phase cultures using the CTAB protocol [48]. Libraries for Illumina sequencing were prepared following the transposon-mediated Nextera XT library preparation kit (Illumina, San Diego, CA, USA) and paired-end sequenced with a 2 × 150 protocol in an Illumina NextSeq 500 system. Comparison to reference sequence of *S. pneumoniae* R6 genome was performed using Newbler 3.0 (Roche, Basel, Switzerland) with fastq files from Illumina used as input sequences. Only changes with total variation percentage ≥ 90% and a total depth ≥ 20 were considered.

### 4.4. Gene and Protein Analysis

Nucleotide sequences were aligned using Clone Manager Suite 7. Protein alignment was performed with Clustal Omega. The RNA stem-loop structure was predicted using the RNA-fold web server (http://rna.tbi.univie.ac.at/cgi-bin/RNAWebSuite/RNAfold.cgi (accessed on 11 November 2022)).

### 4.5. PCR for Bacterial Transformation

PCR fragments for transformation of R6 were amplified using 2 units of Phusion Hot Start II DNA polymerase (Thermo Fisher, Waltham, MA, USA), 90 ng of SPN1852 genomic DNA, 200 μM of each dNTP and 0.5 μM of each primer (listed in Table 2) in a 50 μL final volume. Amplification was performed in a 3-step protocol with an initial denaturation step of 30 s at 98 °C, followed by 35 cycles of 10 s at 98 °C, 20 s at the optimal Tm for each primer pair and 30 s per kb at 72 °C, and a final extension of 10 min at 72 °C. Optimal annealing temperature for each primer pair was estimated using the website https://www.thermofisher.com/tmcalculator (accessed on 11 November 2022), as recommended by manufacturers. PCR fragments were purified from 0.8% agarose gel using NZYGelpure (Nzytech, Lisboa, Portugal) and quantified, prior to the transformation.

### 4.6. MIC Determination

MICs were determined by agar-dilution in Muller-Hinton agar plates supplemented with 5% defibrinated sheep blood according to the CLSI guidelines [50]. The efflux inhibitor reserpine (Res) [17,24] was used at a final concentration of 20 μg/mL, which has been demonstrated to inhibit ethidium bromide, acriflavine, and CPX efflux in the streptococci of the mitis group [31].

### 4.7. Intracellular Fluoroquinolone Accumulation Measurement

Accumulation of CPX on each strain was measured by fluorescence essentially, as previously described [51]. Briefly, a 100 mL culture was grown at 37 °C in A + SY medium up to an OD_620_ = 0.5 and concentrated 20-fold in 0.1 M sodium phosphate with a pH of 7.0. The suspension was equilibrated for 10 min at 37 °C either in the presence or in the absence of 20 μg/mL Res. At this point, the 10 μL sample was taken and serial dilutions were plated in blood–agar plates for further colony counting. Different concentrations of the antibiotic were added to 0.5 mL of the sample and suspensions were incubated at 37 °C for 5 min. Reaction was stopped by adding 2.5 mL of ice-cold 0.1 M sodium phosphate and immediate centrifugation. Cell pellet was suspended with 1 mL of 0.1 M glycine pH 3.0 and incubated for 16 h at room temperature to achieve cell lysis. Samples were centrifuged and antibiotic concentration in the supernatant was measured by fluorescence spectroscopy at 279 nm (excitation wavelength) and 447 nm (emission wavelength). Fluorescence was compared to a standard curve for each antibiotic and accumulation data were converted into ng of CPX per CFU.

### 4.8. RNA Isolation and Quantitative Real-Time PCR

Overnight cultures of R6 were diluted in pre-warmed A + SY up to a final OD_650_ = 0.015, and incubated at 37 °C until OD_650_ ≈ 0.2. Five mL were withdrawn and incubated for an additional hour with 30× MIC of CPX for each strain, or without antibiotic. Cells were harvested by centrifugation (10 min, 8000× *g*, 4 °C); the pellet was immediately immersed in liquid nitrogen and stored at −80 °C until needed. RNA extraction was performed using RNeasy Mini-Kit and DNA was removed through on-column digestion with RNase-Free DNase Set (both from Qiagen). RNA integrity was evaluated by gel electrophoresis and its concentration determined using a Nanodrop 1000 spectrophotometer (Nanodrop Technologies). Reverse transcription reactions were performed with the SuperScript III First-Strand Synthesis System for RT-qPCR (Invitrogen) using 2 µg of total RNA and 50 ng/µL of random hexamers, as previously described [52]. A mock cDNA synthesis reaction was performed, with the reverse transcriptase replaced by water. The primer efficiency was determined (≥98%) and quantitative PCR reactions were performed, as previously described [53] in the LightCycler ^®^ 480 System (Roche Applied Science) using 1 µL of cDNA, 0.25 µM of each primer (listed in Table 2) and 5 µL of SsoAdvanced Universal SYBR Green Supermix (BIO-RAD) in a 10 µL final volume. Relative changes in the gene expression were calculated by the 2^−∆∆Ct^ method, as previously described [54] using the 16S rRNA as an internal reference gene. Two technical and, at least, three biological replicates were performed.

## Figures and Tables

**Figure 1 antibiotics-11-01837-f001:**
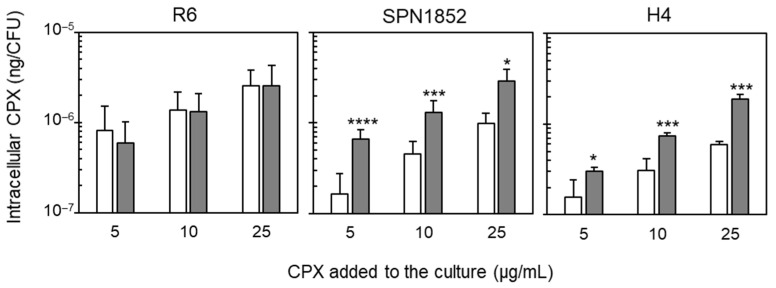
Effect of Res on intracellular CPX-accumulation in SPN strains. Fluorometric measurement of intracellular accumulation of CPX in the indicated strains after 5 min incubation with the specified CPX concentrations either in the presence (dark gray bars) or in the absence (white bars) of 20 µg/mL Res. At least three independent replicates were conducted and mean values with and without Res of each strain were statistically compared using Student’s *t*-test (* *p* < 0.05; *** *p* < 0.005; **** *p* < 0.001).

**Figure 2 antibiotics-11-01837-f002:**
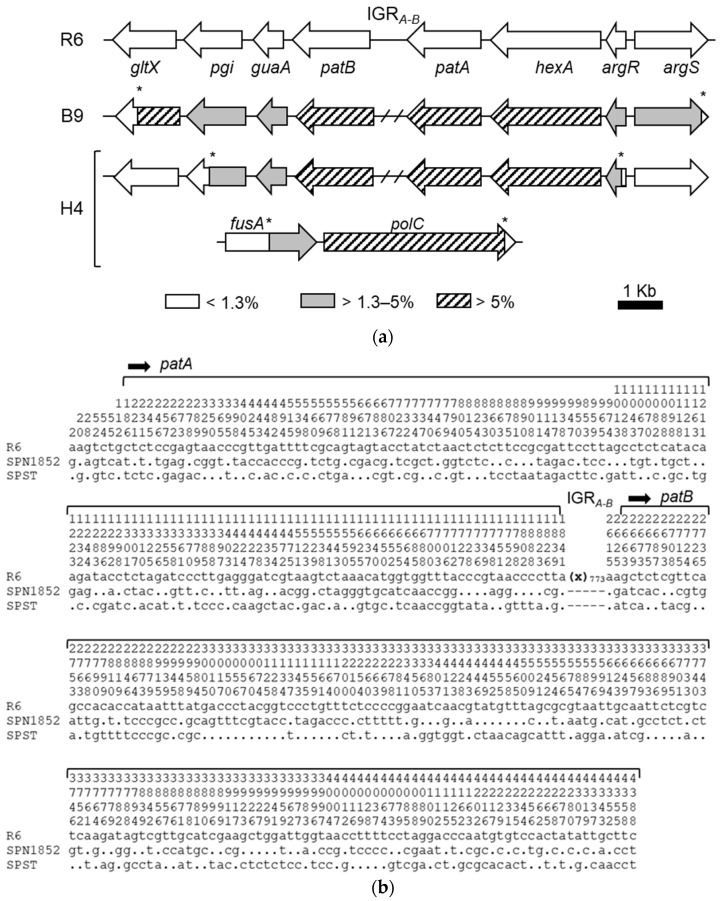
Genomic map of recombination regions and *patAB* sequence comparison. (**a**) Schematic representation of the regions in B9 and H4 compared to R6. Nucleotide divergence with respect to R6 in the genes (arrows) is indicated. Asterisks denote the recombination points on each transformant. The *patA*-*patB* intergenic region (IGR*_A-B_*) is labelled in R6 and its absence in SPN1852 and SPS type strain (SPST) is illustrated by a crosshatched line. (**b**) Comparison of *patAB* operon nucleotide sequence between R6, SPN1852, and SPST (NCBI reference sequence: NC_015875.1). Sequence starts at the first nucleotide after the stop codon of the upstream *hexA* gene. Nucleotide positions are indicated vertically above the sequence. Only differences with respect to R6 are shown. Nucleotides identical to R6 are illustrated by a dot and absent sites are represented by a dash. Sequences of H4 and B9 were identical to that of SPN1852. IGR between *patA* and *patB* (IGR*_A-B_*) is represented by (x) followed by the number of nucleotides not included in the sequence.

**Figure 3 antibiotics-11-01837-f003:**
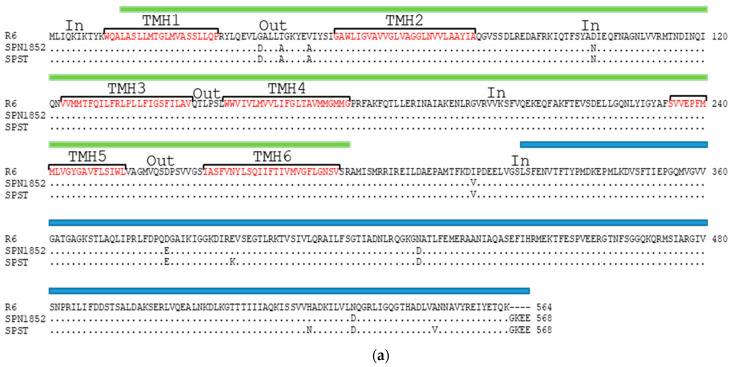
Comparison of PatA (**a**) and PatB (**b**). Multiple sequence alignment of proteins from R6, SPN1852 and SPST. R6 was taken as reference. Identical residues in SPN1852 and SPS^T^ are illustrated by dots and only residues differing from R6 are shown. Predicted domains using ScanProsite (https://prosite.expasy.org/scanprosite/ (accessed on 22 August 2022)): transmembrane domain (green) and ATP-binding domain (blue). Transmembrane alpha-helixes (TMH) predicted by https://dtu.biolib.com/DeepTMHMM (accessed on 14 November 2022) are labeled in red and numbered from 1 to 6 in PatA, and from 7 to 12 in PatB. Extracellular (Out) and intracellular (In) loops are indicated.

**Figure 4 antibiotics-11-01837-f004:**
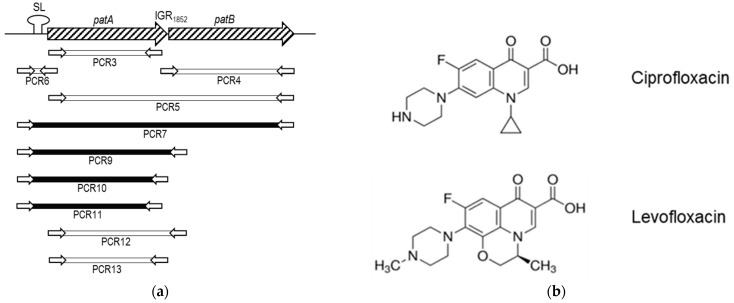
R6 CPX-resistant transformants obtained with PCR fragments amplified from SPN1852. (**a**) Scheme of the *patAB* operon of SPN1852 and its upstream region, and the PCR amplimers used. Genes are represented by dashed arrows and the stem-loop (SL) in the upstream region is illustrated. PCR fragments used in transformation (rectangles) and the corresponding primers (white arrows, not drawn to scale) are shown. Those rendering CPX-resistant transformants are depicted in black; (**b**) chemical structures of ciprofloxacin and levofloxacin; (**c**) table showing the MICs for transformants of LVX and CPX in the presence (+R) or absence of Res. PCR fragments used as donors (1) and clone numbers (2) are indicated. SPN1852 amino acid changes present in PatA and PatB proteins of each transformant are shown.

**Figure 5 antibiotics-11-01837-f005:**
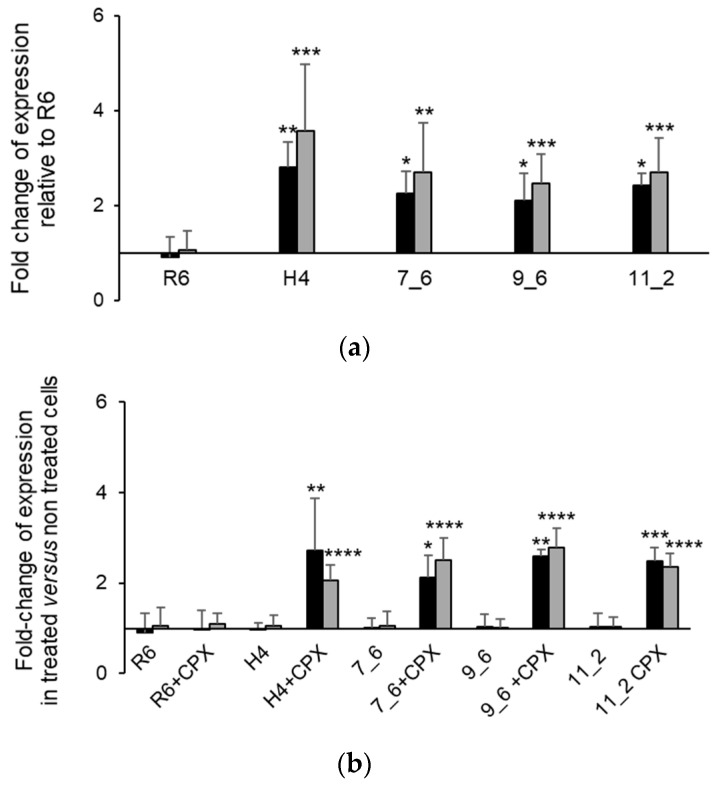
CPX induces *patAB* expression in SPN. Strains were cultured until an OD_620_ = 0.2 and treated with 30 × MIC of CPX or non-treated for 1 h. MIC values considered for each strain were those detailed in Figure 4b. Total RNA was extracted and used in RT-qPCR. Fold-change of *patA* (black) and *patB* (gray) transcript levels were determined using the ΔΔCT method. (**a**) Transcript levels on each strain relative to R6 in the absence of CPX; (**b**) transcript levels in treated *versus* non-treated cultures for each strain. Values shown are the mean of at least three independent replicates (±SD). Statistical significance relative to the reference condition on each panel was tested by the *t*-test with significant data points being highlighted by asterisks (* *p* < 0.05; ** *p* < 0.01; *** *p* < 0.001; **** *p* < 0.0001).

**Figure 6 antibiotics-11-01837-f006:**
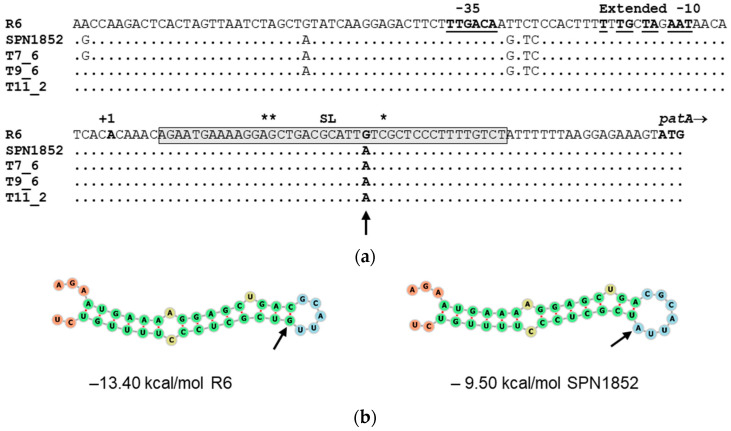
Sequence features of *patAB* upstream region. (**a**) Nucleotide sequence alignment of the intergenic region between *hexA* and *patA* of strains R6, SPN1852, and the indicated PCR transformants. R6 sequence is shown. Identical nucleotides are illustrated by a dot and only nucleotide differences are displayed. The putative −35 and extended −10 sequences of the promoter are underlined and the start codon (ATG) of *patA* is shown in boldface. The transcriptional start site (+1) identified previously [32] is indicated in bold, and the nucleotide sequence of the SL as predicted by RNA-fold is delimited by a gray rectangle. Asterisks denote location of mutations identified previously at the SL; (**b**) diagram showing the predicted secondary structure at the 5’-end of *patA* transcript of SPN1852 and SPNR6 strains by RNA-fold minimum free energy prediction. Bases are colored by structure according to the following scheme; green: stems, yellow: interior loops, blue: hairpin loops and orange: 5’ and 3’ unpaired region. The corresponding free energies are shown. G + 31→A mutation is labelled by an arrow.

**Table 1 antibiotics-11-01837-t001:** Susceptibilities to FQs of SPN chromosomal transformants determined by blood–agar plate in the presence or in the absence of 20 µg/mL of Res, compared to R6 and SPN1852.

Strain	MIC (µg/mL) ^1^			
	LVX	LVX + Res	CPX	CPX + Res
R6	1	1	0.5	0.5
SPN1852	4	1	8	0.5
A1	4	1	8	0.5
A3	4	1	8	0.5
A4	4	1	8	0.5
A5	4	1	8	0.5
H4	4	1	8	0.5
B9	4	1	8	0.5
D1	4	1	8	0.5

^1^ MIC values ≥ 4 µg/mL considered resistant [30]. A decrease of ≥4-fold of MICs in the presence of Res was considered as positive for efflux phenotype.

**Table 2 antibiotics-11-01837-t002:** Oligonucleotides used in this study.

Name	Sequence	Application	Reference
PatBFup	CGCATGCAGACTTGGTTGCCA	PCR4 and sequencing	[32]
PatARACE2	CCAATCAACCAAGCCCCGATAC	PCR6 and sequencing	[32]
HexAF757	CCAAGATTGCTGGCTTGCCAGC	PCRs 6, 7, 9, 10, 11 and sequencing	[32]
PatBRDown	ATGGACAAGAAAAAGCTGCCC	PCRs 4, 5, 7 and sequencing	[32]
patBR3	CTCGTTGGTCGACTCTGCAATC	PCRs 9, 12 and sequencing	This study
patBR2	TGCCAAAAAAATTGAACTGTCTTC	PCRs 10, 13 and sequencing	This study
PatARDown	TGGCAACCAAGTCTGCATGCG	PCRs 3, 11 and sequencing	[32]
PatAF17	GATGACAGGCTTGATGGTTGC	PCRs 3, 5, 12, 13 and sequencing	[32]
PatARTR	AACGACTAGATTTCCCGCAT	Sequencing and qRT-PCR	[32]
PatAF1	GCTAGAATAACATCACACAAACAG	sequencing	This study
PatAF2	AAGCCAGATTATCTTTACCATTG	sequencing	This study
PatAF3	GTGAATTCATTCATCGTATGGAG	sequencing	This study
PatAR1	CATTGGATAGGTAAAGGTCAC	sequencing	This study
PatAR2	AACAATCACCCACCACAGAG	sequencing	This study
PatBF2	AACGTAGATACGGTGACAGAAAGC	sequencing	This study
PatBRTR	GCACGCGCTCATTTTGTTCA	sequencing	[32]
PatBRTF	GCACCCCATTGGCTTTCCTTA	sequencing	[32]
PatARTF2	GCTTGGTTGATTGGTGTGGC	qRT-PCR	This studty
patBRTF2	AATACACCAACCTCCAGCAG	qRT-PCR	This studty
16SF	AGCGTTGTCCGGATTTATTG	qRT-PCR	[49]
16SR	CATTTCACCGCTACACATGG	qRT-PCR	[49]

## Data Availability

Illumina fastq files from H4 and B9 strains of *S. pneumoniae* were deposited in the NCBI’s Sequence Read Archive (accession number PRJNA903760) and can be found at https://www.ncbi.nlm.nih.gov/sra/PRJNA903760 (accessed on 22 November 2022). Nucleotide sequence of *patAB* operon from SPN1852 (OP903390), H4 (OP903391) and B9 (OP903392) obtained by the Sanger method were deposited in the NCBI’s GenBank.

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
