# Peer review of "Role of PatAB Transporter in Efflux of Levofloxacin in Streptococcus pneumoniae"

_antibiotics, 2022, doi:10.3390/antibiotics11121837_

Round 1
Reviewer 1 Report
The paper by Amblar et al reports analysis of the role of PatAB in efflux of fluoroquinolones in a S. pneumoniae strain. The authors identify a mutation in a stem-loop structure upstream of the patAB operon that increases expression of the efflux pump, and mutations in PatA that contribute to resistance. In my opinion the paper is well-written and consistent, results are presented and discussed clearly.
Minor points that should be addressed by the authors:
1) the legend to figure 2B is unclear, please define what the numbers above the nucleotide sequence represent
2) when discussing CPX vs LVX efflux/resistance (lines 357-371) it may be helpful to add a figure with the structure of CPX and LVX to highlight differences between the two antibiotics
3) line 161: reference is to figure 2a rather than 2b?
4) line 401: 'condition' not 'consition', and please check some other typos in results and discussion sections
Author Response
The paper by Amblar et al reports analysis of the role of PatAB in efflux of fluoroquinolones in a S. pneumoniae strain. The authors identify a mutation in a stem-loop structure upstream of the patAB operon that increases expression of the efflux pump, and mutations in PatA that contribute to resistance. In my opinion the paper is well-written and consistent, results are presented and discussed clearly.
Minor points that should be addressed by the authors:
Point 1. the legend to figure 2B is unclear, please define what the numbers above the nucleotide sequence represent
Response 1. We thank the referee for pointing out this issue. Indeed, there was a mistake in legend of Figure 2 and the paragraph describing sequence in panel (b) was placed in description of panel (a). We apologize for the mistake and now the whole paragraph has been moved to the correct place.
Point 2. when discussing CPX vs LVX efflux/resistance (lines 357-371) it may be helpful to add a figure with the structure of CPX and LVX to highlight differences between the two antibiotics
Response 2. Following the referee suggestion, a scheme of the chemical structure of both CPX and LVX has been included in Figure 4 as a new panel (b). We thank the referee for his/her suggestion.
Point 3. line 161: reference is to figure 2a rather than 2b?
Response 3. We believe that both panels of figure 2 should be referred at this point. Panel (a) shows the range of percentage of divergence between H4 and B9, relative to R6, but the exact nucleotide divergence shown by SPN1852 (which is identical to H4 and B9) can only be estimated from the scheme shown in panel (b). Therefore, we have changed the reference to Figure 2 without panel specification.
Point 4 line 401: 'condition' not 'consition', and please check some other typos in results and discussion sections
Response 4. We thank the referee for noticing this typo error
Please note that, since all reviewers agree in recommending minor revisions in terms of English, the manuscript has been checked by a UK-based colleague. Typos have been corrected throughout and a number of sentences have been slightly modified in order to improve the text and the ease to read it.
Reviewer 2 Report
In this work “Role of PatAB transporter in efflux of levofloxacin in Streptococcus pneumoniae"), Amblar, et al. investigate the fluoroquinolones resistance of a clinical strain of S. pneumoniae. The authors’ determine that the resistance is due to the PatAB efflux pump, investigate the mechanism of a regulatory changes caused by a mutation in a non-coding region, and find differences in response to two different fluoroquinolones.
This paper, which is interesting and well written, should be of interest those interested in antibiotic resistance and efflux. However, there are several issues with data analysis and presentation that need to be addressed to strengthen the paper.
Comments
1. Ln 20-21, 77-78: These sentences say that the promoter region and four mutations in patA are sufficient for resistance. Although this is true, it implies that the mutations been individually tested. Please change these sentences to make it clear that the region containing these mutation is sufficient.
2. Table 1: Since, R6 and SPN1852 are compared before the transformants are mentioned it would be helpful to move SPN1852 to the second row of the table.
3. Figure 1: It appears that the –Res samples are being statistically compared to the +Res samples but this is not specified. Please add information on the samples being compared in the figure legend.
4. Figure 5: It is not clear what strains are being compared with the t-tests. Please specify in the figure legend.
Author Response
Point 1. Ln 20-21, 77-78: These sentences say that the promoter region and four mutations in patA are sufficient for resistance. Although this is true, it implies that the mutations been individually tested. Please change these sentences to make it clear that the region containing these mutation is sufficient.
Response 1. Following the referee suggestion, both sentences has been changed accordingly
Point 2. Table 1: Since, R6 and SPN1852 are compared before the transformants are mentioned it would be helpful to move SPN1852 to the second row of the table.
Response 2. We agree with the referee that moving up the SPN1852 data to the second row in Table 1 may help comparison and we have changed accordingly
Point 3. Figure 1: It appears that the –Res samples are being statistically compared to the +Res samples but this is not specified. Please add information on the samples being compared in the figure legend.
Response 3. Following the referee suggestion, statistical comparison of -Res and +Res is now specified in the legend of Figure 1
Point 4. Figure 5: It is not clear what strains are being compared with the t-tests. Please specify in the figure legend.
Response 4. Figure 5 shows fold-change of patA and patB transcript levels relative to that of R6 in the absence of CPX (in panel a) and in treated versus non-treated cultures for each strain (in panel b), as indicated in the figure legend, and so the statistical analysis. The ΔΔCT method for calculation of gene expression is always presented as fold-change relative to a reference condition, which is the same condition used for statistical comparison. Nevertheless, attending the referee suggestion we have added this information in the figure legend in order to make it more clear.
Please note that, since all reviewers agree in recommending minor revisions in terms of English, the manuscript has been checked by a UK-based colleague. Typos have been corrected throughout and a number of sentences have been slightly modified in order to improve the text and the ease to read it.
Author Response
We deeply appreciate the positive evaluation of our manuscript by the referee. Please note that, since all reviewers agree in recommending minor revisions in terms of English, the manuscript has been checked by a UK-based colleague. Typos have been corrected throughout and a number of sentences have been slightly modified in order to improve the text and the ease to read it.